Randomness analysis of end-to-end delay in random forwarding networks

Wang Xiaowen 1
Huang Jie 1 2 jhuang@seu.edu.cn
Duan Zhenyu 1
http://orcid.org/0000-0002-6778-3674 Xu Yao 1
Yao Yifei 1
1 School of Cyber Science and Engineering, Southeast University , Nanjing, Jiangsu , China
2 Purple Mountain Laboratories , Nanjing, Jiangsu , China
Jhanjhi Noor
Electronic publication date: 2022 Apr 6
Publication date: 2022
Volume: 8
Electronic Location ID: e942
Received 2021 Sep 30; Accepted 2022 Mar 14
Copyright: © 2022 Wang et al.
Copyright year: 2022
Copyright holder: Wang et al.
License: This is an open access article distributed under the terms of the Creative Commons Attribution License, which permits unrestricted use, distribution, reproduction and adaptation in any medium and for any purpose provided that it is properly attributed. For attribution, the original author(s), title, publication source (PeerJ Computer Science) and either DOI or URL of the article must be cited.
License URL: https://creativecommons.org/licenses/by/4.0/

Keywords: Random forwarding networks, End-to-end delay, Randomness analysis, Symbol matrix, Key generation

Funding: National Key Research and Development Program of China 2018YFB2100400 This work was supported by the National Key Research and Development Program of China (No. 2018YFB2100400). The funders had no role in study design, data collection and analysis, decision to publish, or preparation of the manuscript.

==============================
Random forwarding networks play a significant role in solving security and load balancing problems. As a random quantity easily obtained by both sender and receiver, the end-to-end delay of random forwarding networks can be utilized as an effective random source for cryptography-related applications. In this paper, we propose a mathematical model of Random forwarding networks and give the calculation method of end-to-end delay distribution. In exploring the upper limit of the randomness of end-to-end delay, we find that the end-to-end delay collision of different forwarding routes is the main reason for the decrease of end-to-end delay randomness. Some of these collisions can be optimized by better network deployment, while others are caused by some interesting network topology, which is unavoidable. For further analysis, we propose an algorithm to calculate the inevitable collision in random forwarding networks skillfully by using Symbol Matrix, and we give the optimal node forwarding strategy with the maximum randomness of the end-to-end delay for a given number of middle forwarding nodes and forwarding times. Finally, we introduce a specific application of generating symmetric keys by using the randomness of the end-to-end delay.

Introduction

A drunk man left a bar to go home. When he arrived at a crossroads, he couldn’t recognize the way back because of drunkenness. There were two choices in front of him. One choice is to stay in place for a while, and the other choice is to choose a road in front of him at random. The streets of this city extend in all directions, and the drunk man could go anywhere he could go. After walking a few blocks, the drunk man woke up and went back to his home directly. Because the drunk man goes to the bar every day, his wife at home is curious about the regularity of the time he returns.

The problem of the drunk man returning home can be modeled by random forwarding networks and the time drunk man spends on the road is the end-to-end delay of a random forwarding route. Suppose we have a random forwarding network G consisting of m middle forwarding nodes Z1, Z2,…, Zm. G plays the role of forwarding the delay measurement data packet sent by Alice to Bob. The forwarding rules are as follows: Firstly, Alice randomly selects a middle forwarding node to send the initial delay measurement data packet.

Secondly, this middle forwarding node randomly selects other middle forwarding nodes as the next hop of forwarding or forwards the packet to itself, and stipulates that the total forwarding times of the delay measurement data packet is N, which is recorded in the data packet. Every time the data packet is forwarded, the remaining forwarding times is reduced by 1 by the currently receiving middle forwarding node.

Finally, when the remaining forwarding time becomes 0, the current middle forwarding node directly forwards this delay measurement data packet to Bob and finishes this forwarding.

Obviously, the end-to-end delay is related to the number of middle forwarding nodes, random forwarding strategy, and forwarding times. The main content of this paper is to reveal this relationship.

First, let us introduce the definition of random forwarding networks. Random forwarding networks (RFNs) are networks consisting of several network nodes called middle forwarding nodes with random forwarding as the forwarding strategy. Different from the Open Shortest Path First (OSPF) forwarding strategy, RFNs do not focus on efficient data transmission, but on the security application and load balancing in the process of data forwarding.

A security application is embodied in the attacker’s inability to track the data in the Random forwarding networks because the forwarding node is randomly selected rather than determined by some forwarding rules (Duan, Al-Shaer & Jafarian, 2013). The famous Tor network takes advantage of the anonymity of random forwarding, and Tor agents replace users to visit service sites to keep users secure. Using onion routing technology, access requests are randomly forwarded among several Tor network agents, hiding users’ real addresses (Syverson et al., 2001). In optical transport networks (OTN), random forwarding is potentially more secure than explicit forwarding, and the probability that a wiretapper recovers a whole secure data as the first try is in the range of 10−7 (Engelmann, Zhao & Jukan, 2014).

While load balancing can evenly distribute tasks to multiple working nodes, which is an essential technology in high-performance web services (Liu, Jin & Yang, 2013). In wireless sensor networks (WSNs), random forwarding can provide a more stable and longer lifetime of networks (Li & Kim, 2015). In addition, RFNs are strongly extensible because the random forwarding strategy makes every node have equal status, and it can flexibly add new nodes without changing the basic forwarding logic. Because of the flexibility, RFNs also has strong robustness. When an abnormal node in an RFN is detected, the whole RFN can still work effectively by deleting the abnormal node from the forwarding list.

The whole network delay from Alice to Bob is the end-to-end delay. In RFNs, the end-to-end delay has strong randomness, and it can be easily measured by both sender and receiver, which is of great significance in cryptography (Abdelkefi & Jiang, 2011). The delay between middle forwarding nodes has stability and reciprocity, in which stability means that there is no obvious fluctuation in the delay between middle forwarding nodes within a short period time (within a few minutes), while reciprocity means that the communication round-trip delay is approximately equal (Choi et al., 2004).

In physical layer security, it is a valuable technology to generate security keys by using the reciprocity and randomness of wireless channels, which can enable both parties to quickly establish a secure communication channel (Sánchez et al., 2020). The lightweight security solutions relying on key generation from wireless channels are eminently suitable for the Internet of Things (IoTs) (Zhang et al., 2020). Similarly, the end-to-end delay with reciprocity and randomness in RFNs can also be used to achieve the same purpose. However, the difference is that using wireless channel characteristics to generate keys has great restrictions on communication distance while using network characteristics has no such restrictions, which can achieve cross-regional key negotiation.

Therefore, in order to further explore the potential of RFNs in multi-node cross-domain secret sharing and key distribution, this paper mainly discusses the randomness of end-to-end delay in RFNs. The main contributions of this paper are summarized as follows: We propose the mathematical model of RFNs and derive the mathematical formula of the end-to-end delay distribution.

We present a quantitative calculation method of the end-to-end delay randomness based on information entropy and give a theoretical explanation.

We explore the forwarding strategy that maximizes the randomness of end-to-end delay when the number of middle forwarding nodes and forwarding times is constant. We reveal that the main reason for the decrease of the randomness of end-to-end delay is delay collision and provide the optimal forwarding strategy and the theoretical upper limit of end-to-end delay randomness under different numbers of middle forwarding nodes and random forwarding times.

We introduce the application of cross-domain key distribution using the randomness and reciprocity of end-to-end delay.

Rfns model

The end-to-end delay probability distribution

In this section, we will first give the algebraic relationship between end-to-end delay distribution and the forwarding strategy of middle forwarding nodes.

Because the time delay between any two nodes in the forwarding network is stable in a short term, once the deployment of forwarding network G is completed, the time delay between nodes is determined in this short term. Here are some symbol habits used in this paper, the delay and the forwarding probability between Alice and middle forwarding nodes Zi are denoted as dai and pai respectively, the delay and the forwarding probability between middle forwarding nodes Zi and middle forwarding nodes Zj are denoted as dij and pij respectively, and the delay between Bob and middle forwarding nodes Zi are denoted as dib.

In the process of forwarding, we use the delay monomial pxd keep the cumulative information of probability and the cumulative information of delay because of such property: p1xd1⋅p2xd2=p1p2xd1+d2. Take Fig. 1 as an example, the delay and the probability of Alice→Z1→Z2→Z3→Bob for the forwarding route r can be calculated as

Figure 1 Random forwarding networks for m = 3.

prxdr=∏pixdi=(pa1p12p23p3b)x(da1+d12+d23+d3b)

Figure 2 describes all possible forwarding routes from Alice to Bob under general conditions of m nodes and N times and defines the set of these routes as S. Since each route corresponds to a delay monomial prxdr, then the distribution of end-to-end delay is the sum of the delay multinomial prxdr corresponding to all routes, and we express this sum as

Figure 2 All possible forwarding routes of m nodes for N times.

p(x)=∑r∈Sprxdr

After combining like terms, we get p(x)=∑i=1npixdi, which means that the probability of taking di as the end-to-end delay is pi.

p(x) is the polynomial form of the end-to-end delay probability distribution. Considering the multi-layer network structure of forwarding routes, the vector form of the end-to-end delay distribution polynomial p(x) can be calculated as follows

(1) p(x)=sTPNt

where s=(pa1xda1pa2xda2…pamxdam)T is the initial forwarding vector forwarded by Alice to the middle forwarding nodes, t=(xd1bxd2b…xdmb)T is the end forwarding vector forwarded by middle forwarding nodes to Bob, and P is the forwarding matrix of middle forwarding nodes forwarding to each other, that is

P=(p11xd11…p1mxd1m⋮⋱⋮pm1xdm1…pmmxdmm)

When the forwarding network is deployed, the delay dij between any two nodes is determined. According to Eq. (1), p(x) is uniquely determined by the random forwarding strategy. Let PA=(pa1pa2…pam)T denotes the initial random forwarding strategy forwarded by Alice to the middle forwarding nodes. Let PZ denotes the random forwarding strategy of middle forwarding nodes forwarding to each other, that is

PZ=(p11…p1m⋮⋱⋮pm1…pmm)

Measurement of end-to-end delay randomness

As shown in Fig. 3, the network model of end-to-end delay generated by a random forwarding network is regarded as a black box. Given forwarding strategy (PA, PZ), this black box will randomly generate end-to-end delay data, which will obey the probability distribution defined by p(x). This is similar to a discrete source sending uncertain symbols in communication. The randomness of a source sending symbols can be measured by information entropy, which reflects the uncertainty of a source by calculating the average self-information of symbols (Shannon, 1948).

Figure 3 End-to-end delay generation model.

Therefore, by calculating the information entropy of end-to-end delay, we can quantitatively analyze its randomness. If the randomness of end-to-end delay is exploited to generate the secret key, the effective length of the secret key is proportional to the randomness of end-to-end delay. For example, if the end-to-end delay is given by the front and back of a coin thrown, d1 will be generated on the front side and d2 will be generated on the backside, that is to say, the end-to-end delay will only generate two possible values with the same probability, so there are at most two corresponding secret keys. Although the secret key length can be expanded by some algorithms like Hash (Bellare, Canetti & Krawczyk, 1996), the effective key code length is actually only 1 bit, which is the information entropy of the end-to-end delay.

The measurement formula of end-to-end delay randomness is as follows

(2) Hd=−∑ipilog⁡pi

where pi are the coefficients of p(x) calculated by Eq. (1).

Optimization of the randomness of end-to-end delay

This section mainly discusses how to improve the randomness of end-to-end delay, which is of great significance in cryptography.

Evitable collision and inevitable collision of end-to-end delay

End-to-end delay collision (hereinafter referred to as collision) means that two different forwarding routes have the same end-to-end delay. Collision is one of the main reasons leading to the decrease of the randomness of end-to-end delay because of the reduction of end-to-end delay sample space.

Collisions that can be solved by adjusting RFNs deployment are referred to as evitable collisions. Otherwise, they are referred to as inevitable collision. These two collisions are described in detail below.

Evitable collision

In order to show this collision intuitively, an example as shown in Fig. 4 is provided, which is an equal delay forwarding network with two middle forwarding nodes, in which the delay between any two nodes is approximately the same (replaced by 1).

Figure 4 Equal delay forwarding network for m = 2.

Taking single forwarding as an example, it is easy to find from Fig. 4 that the end-to-end delay of route Alice→Z1→Z1→Bob is the same as Alice→Z2→Z2→Bob, and the end-to-end delay of route Alice→Z1→Z2→Bob is the same as Alice→Z2→Z1→Bob, that is to say, the end-to-end delays of these two pairs of routes collide.

The end-to-end delay distribution polynomial corresponding to Fig. 4 is

p(x)=(pa1xpa2x)T(p11p12xp21xp22)(xx)=pa1p11x2+pa2p22x2+pa1p12x3+pa2p21x3

The collision of end-to-end delay is reflected by the existence of like terms in the end-to-end delay distribution polynomial, and the existence of like terms reduces the randomness of end-to-end delay. For example, Alice→Z1→Z1→Bob corresponds to pa1p11x2, Alice→Z2→Z2→Bob corresponds to pa2p22x2, which are like terms.

Equal delay forwarding networks are prone to delay collisions. To avoid such collisions, the deployment of forwarding networks can be adjusted, such as the forwarding network shown in Fig. 5.

Figure 5 Adjust the deployed forwarding network for m = 2.

Similarly, taking a single forwarding as an example, the corresponding end-to-end delay distribution polynomial is

p(x)=(pa1xpa2x2)T(p11p12x3p21x3p22)(xx3)=pa1p11x2+pa2p22x5+pa1p12x7+pa2p21x6

There is no like term in the adjusted end-to-end delay distribution polynomial, that is to say, the end-to-end delay corresponding to each possible forwarding route is different, which improves the randomness of the measurement delay. This kind of collision is called evitable collision.

Inevitable collision

Taking m = 2 and N = 2 as an example, the end-to-end delay distribution polynomial is as follows

p(x)=sTP2t=sT(p11p12xdp21xdp22)2t=sT(p112+p12p21x2dp11p12xd+p12p22xdp21p11xd+p22p21xdp222+p21p12x2d)t

It can be found that the internal elements of matrix P2 have like terms, such as p11p12xd + p12p22xd in the second column of the first row and p21p11xd + p22p21xd in the first column of the second row, which will lead to the existence of like terms in the expansion. The collision caused by such like terms can not be avoided by adjusting the deployment. So, we call this kind of collision inevitable collision.

Taking the forwarding network in Fig. 5 as an example, make a forwarding route map under two forwarding, which is shown as Fig. 6. The blue route (-··) is Alice→Z1→Z1→Z2→Bob and the yellow route (-·) is Alice→Z1→Z2→Z2→Bob, which correspond to p11p12xd and p12p22xd from the second column of the first row in matrix P2 respectively. Since the two routes share all the edges that can be changed by deployment, they are bound to collide.

Figure 6 Inevitable collision of end-to-end delay (take forwarding network in Fig. 5 as an example).

Fast calculation of inevitable collision using symbol matrix

The collision of end-to-end delay is the main reason for the decrease of the randomness of end-to-end delay. The evitable collision can be solved by adjusting the deployment, while the inevitable collision is an unavoidable problem in the optimization of the randomness of end-to-end delay. Therefore, this subsection introduces a method for quickly calculating the inevitable collision in RFNs.

We have known that the inevitable collision depends on whether there are like terms in the internal elements of matrix PN, which is an inherent property of matrix power operation and is independent of the value of the specific elements of the matrix itself. Symbol matrix is a matrix composed of simple symbols, which is very suitable for revealing the structure of like terms in the internal elements of matrix PN.

The diagonals of the symbol matrix are all replaced by 1, which represents that the nodes forward to themselves will not change the end-to-end delay. The non-diagonals represent the delay between different nodes and are replaced by symbols. In fact, the symbol matrix is only a simplification of the forwarding matrix P. In this paper, Sm is used to denote the symbol matrix of the forwarding matrix P with m nodes. Note that Sm is symmetric.

For example, the symbol matrix S2 for m = 2 is

S2=(1aa1)←(p11p12xdp21xdp22)

If N = 2, the symbol matrix S22 is

S22=(1+a22a2a1+a2)←(p112+p12p21x2dp11p12xd+p12p22xdp21p11xd+p22p21xdp222+p21p12x2d)

where 2a is the result of merging like terms, the coefficient represents that the number of inevitable collision routes is 2.

With the help of symbol matrix, it is easier to calculate the inevitable collision in complex cases. Taking m = 3 as an example, the symbol matrix S3 is

S3=(1aba1cbc1)

When N = 2, the symbol matrix S32 is

S32=(1+a2+b22a+bc2b+ac2a+bc1+a2+c22c+ab2b+ac2c+ab1+b2+c2)

We find that the form of the elements on the main diagonal of S32 is consistent, and the form of the elements on the upper triangle and the lower triangle (except the main diagonal) of S32 is consistent. The difference only exists in the rotation of symbols, which is called rotation consistency. That is to say, as long as the first two elements of the first line of S32 are calculated, the remaining elements can be recovered by rotation consistency.

So S32 can be compressed as

S32=(1+a2+b2,2a+bc)a,b,c

Where the elements in () is the first two elements in S32 and the subscript a,b,c denote the symbols of rotation.

Two operators are used to recover the original S32 from the compressed S32. The first operator is the cyclic permutation transformation R:

(f1(a,b,c)f2(a,b,c)f3(a,b,c))→R⁡(f3(σ(a,b,c))f1(σ(a,b,c))f2(σ(a,b,c)))

where permutation operator σ=(abccab) and it makes

S32=(μR(μ)R2(μ)),μ=(1+a2+b22a+bc2b+ac)

The second operator is replacement transformation eij:

fi(a,b,c)→eij⁡fj(a,b,c)=fi(eij(a,b,c)),i,j≥2

where eij can be generated by Sm[:,j] = eij(Sm[:,i]). Sm[:,i] denotes the ith column of Sm.

In recovering the compressed S32, we need e23=(abba)=a↔b to recover μ as

μ=(f1(a,b,c)f2(a,b,c)f2(e23(a,b,c)))

By using operators R and eij, the complete matrix S32 can be recovered from the first two elements of the S32. This property is universal, and there is such rotation consistency for any number of nodes and any number of forwarding times (See APPENDIX for proof).

Now, we will show how to use these two operators to calculate S33 easily:

S3=(1,a)a,b,c,μ0=(1ab),μ1=R(μ0)=(a1c),γ=(1ae23(a))=μ0

S32=(γTμ0,γTμ1)a,b,c=(1+a2+b2,2a+bc)a,b,c,γ=(1+a2+b22a+bce23(2a+bc))=(1+a2+b22a+bc2b+ac)

S33=(γTμ0,γTμ1)a,b,c=(1+3a2+3b2+2abc,a3+ab2+ac2+3a+3bc)a,b,c

where γ is the first column of S3N.

Generally, the fast power of symmetric symbol matrix (FPSSM) is given by Algorithm 1 to calculate matrix SmN easily. Because every loop in FPSSM only needs to calculate two times vector multiplication, the algorithm reduces the time complexity of polynomial matrix multiplication from O(Nm3) to O(Nm) and the space complexity from O(m2) to O(1). The complexity here refers to the complexity of polynomial multiplication, not the complexity of conventional numerical multiplication.

Algorithm 1 FPSSM(m,N)

Input: m: Dimensions of Symbol Matrix Sm; N: Power of Symbol Matrix Multiplication	
1: Sm = Symbol_Matrix_Generate(m)	
2: R = Cyclic_Permutation_Generate(Sm)	
3: eij = Replacement_Generate(Sm)	
4: f,g = Sm[0,0],Sm[0,1]	
5: μ0,μ1 = Sm[:,0],Sm[:,1]	
6: for i in [1,2,…,N−1] do	
7: γ=[f,g,e23(g),e24(g),…,e2m(g)]T	
8: f←γTμ0μ0	
9: g←γTμ1)μ1	
10: γ=[f,g,e23(g),e24(g),…,e2m(g)]T	
11: SmN=[γ,R(γ),…,Rm−1(γ)]	
12: return SmN	

Now we have powerful tools to study the inevitable collision of RFNs in complex conditions. As long as we calculate SmN, all possible inevitable collisions can be obtained. Take m = 3, N = 3 as an example, every term in S33 whose coefficient is not 1 represents an inevitable collision. Figure 7 shows the inevitable collision of 3a2 and 2abc in S33. Among them, the first figure labeled 3a2 shows a kind of inevitable collision caused by self forwarding, while the second figure labeled 2abc shows another kind of inevitable collision caused by symmetry in the forwarding route map. Of course, these two types are not mutually exclusive. There are also inevitable collisions caused by both self-forwarding and symmetry in forwarding route maps with more middle forwarding nodes.

Figure 7 Two typical inevitable collision of end-to-end delay for m = 3, N = 3.

The upper limit of end-to-end delay randomness and the optimal forwarding strategy

In this subsection, we will explore how to formulate random forwarding strategies to achieve the upper limit of end-to-end delay randomness. We have known that the collision of end-to-end delay will lead to the decrease of randomness, so the first step is to adjust the deployment to remove all evitable collisions. In this way, our goal becomes the optimal forwarding strategy under the inevitable collision deployment.

Our optimization problem is that, for a given non-evitable collision random forwarding network G (including Alice and Bob), what is the optimal forwarding strategy to maximize the information entropy of the end-to-end delay? The mathematical form is described as follows

Given:G=(V,E),V={Alice,Z1,Z2,…,Zm,Bob}

Goal:maxPA,PZ⁡Hd=−∑ipilog⁡pi

where pi are the coefficients of p(x) calculated by Eq. (1).

The maximum entropy problem is a convex optimization, and its optimal solution exists and is unique (Boyd & Vandenberghe, 2004), which is the key to solving this optimization problem.

First, let’s define a cyclic shift permutation operator C on the matrix A∈Rm×m as

C:(a11a12…a1na21a22…a2n…………am1am2…amn)→(a22a23…a21a32a33…a31…………a12a13…a11)

In fact, C is a compound operation of cyclic left shift and cyclic upward shift on the matrix, so any element in the matrix is permuted as follows under the transformation of C

aij→C⁡a[i+1]m[j+1]n

where [i + 1]m = (i mod m) + 1 ensures the cyclic property of the shift.

Operator C has the following three important properties: Property 1

Cm(A)=A,A∈Rm×m

Property 2

C(A)C(B)=C(AB),A,B∈Rm×m

Property 3

C(xTAy)=xTAy,A∈Rm×m,x,y∈Rm

Then, rewrite the end-to-end delay distribution polynomial p(x) with Hadamard Product as

(3) p(x)=sTPNt=(PA∘xDA)T(PZ∘xDZ)NxDB

where xDA=(xda1xda2…xdam)T, xDB=(xd1bxd2b…xdmb)T and xDZ=(xdij)m×m. The operator ○ is the Hadamard product operator defined by

(A∘B)ij=(A)ij(B)ij

Because Hd is calculated by pi, which are the coefficients of p(x), and pi is distributed by PATPZN1 according to the end-to-end delay like term, that is to say, Hd is decided by PATPZN1 (The notation 1 represents a vector of ones of appropriate length).

Since the optimization objective is PA and PZ, by cyclic shifting PA and PZ in p(x) using C, we get

C(p(x))=(C(PA)∘xDA)T(C(PZ)∘xDZ))NxDB

According to Property 2 and Property 3, we have

C(PA)TC(PZ)N1=C(PAT)C(PZN)1=C(PATPZN1)=PATPZN1

Therefore,

Hd(PA,PZ)=Hd(C(PA),C(PZ))

It is known from the uniqueness of the optimal solution of convex optimization that

{PA=C(PA)PZ=C(PZ)

Similarly,

{PA=C(PA)=C2(PA)=…=Cm−1(PA)PZ=C(PZ)=C2(PZ)=…=Cm−1(PZ)

That is

{pa1=pa2=…=pam=1mp11=p22=…=pmmp12=p23=…=pm1…p1m=p21=…=pmm−1

In addition, according to the rotation consistency of the SmN, we know that the forwarding object Z2,Z3,…,Zm can rotate for node Z1, that is

p12=p13=…=p1m

Let p11 = p, p12 = q, PA and PZ are updated as

{PA=1m1PZ=(p−q)I+q11T

where I is the identity matrix with ones down the diagonal. In fact, p represents the self-forwarding probability of middle forwarding nodes, and q represents the forwarding probability between middle forwarding nodes.

Substituting back into Eq. (3), we have

(4) p(x)=sTPNt=1mxDAT(((p−q)I+q11T)∘xDZ)NxDB

Then, our optimization goal is simplified as

maxp,q⁡Hd=−∑ipilog⁡pi

s.t.p+(m−1)q=1,0≤p,q≤1

where pi are the coefficients of p(x) calculated by Eq. (4).

This optimization can be solved by the Karush–Kuhn–Tucker (KKT) conditions of Lagrange multiplier method as

(5) {(m−1)∂Hd∂p=∂Hd∂qp+(m−1)q=1

Considering

P=((p−q)I+q11T)∘xDZ=(pqxd12…qxd1mqxd21p…qxd2m…………qxdm1qxdm2…p)

Because xdij = xdji, P is a symmetric symbolic matrix. Algorithm 1 can be used to calculate PN quickly and get the expression of Hd.

Take m = 3, N = 2 as an example, because S32=(1+a2+b2,2a+bc)a,b,c, we get

P2=(p2+q2x2d12+q2x2d13,2pqxd12+q2xd13+d23)xd12,xd13,xd23

Then, Hd for m = 3, N = 2 is calculated by Eq. (2) as

Hd(m=3,N=2)=log⁡3−p2log⁡(p2)−4pqlog⁡(2pq)−4q2log⁡q2

Figure 8 shows the change of Hd(m = 3, N = 2) (bits) with the change of p. It can be clearly seen from the figure that the best p corresponding to the maximum entropy is the position marked by the red dot.

Figure 8 The change of Hd (bits) with the change of p for m = 3, N = 2.

By substituting back into Eq. (5) and simplifying, we have

{(pq+2)logpq=(pq−2)log⁡2p+2q=1

Through Newton’s Method, the optimal forwarding strategy is

{p≈0.265q≈0.3675

Then we know the best p in Fig. 8 is 0.265, and the maximum entropy Hdmax is 4.333 bits.

Similarly, we can calculate the optimal forwarding strategy under other m and N. Some results are given in the Tables 1 and 2. Table 1 provides the p value of the optimal forwarding strategy, which is the probability of self-forwarding. While the probability q representing the forwarding probability between middle forwarding nodes can be calculated by p=1−pm−1. Table 2 provides the maximum entropy Hdmax, which is the upper limit of end-to-end delay randomness. From these two tables, we can find that with the increase of forwarding times N, the p value of the best forwarding strategy tends to be stable gradually and the growth rate of the maximum entropy Hdmax is gradually decreasing, that is to say, it is impossible to increase the end-to-end delay randomness by the unlimited number of forwarding times. When the number of forwarding times cannot increase the end-to-end delay randomness, the only effective way is to add more middle forwarding nodes.

Table 1 Optimal forwarding strategy (p value).

p		N	
	1	2	3	4	5	6	7	8	9	
m	2	0.5	0.5	0.5	0.5	0.5	0.5	0.5	0.5	0.5	
	3	0.333	0.265	0.237	0.231	0.231	0.232	0.233	0.234	0.236	
	4	0.25	0.175	0.146	0.135	0.132	0.132	0.132	0.132	0.132	
	5	0.2	0.13	0.104	0.092	0.088	0.086	0.085	0.085	0.084	
	6	0.167	0.103	0.08	0.069	0.064	0.062	0.06	0.06	0.059	
	7	0.143	0.086	0.065	0.055	0.05	0.047	0.046	0.045	0.044	
	8	0.125	0.073	0.055	0.045	0.041	0.038	0.036	0.035	0.034	
	9	0.111	0.064	0.047	0.039	0.034	0.031	0.03	0.029	0.028	

Table 2 The upper limit of end-to-end delay randomness (bits).

H dmax		N	
	1	2	3	4	5	6	7	8	9	
m	2	2	2.5	2.811	3.03	3.2	3.333	3.447	3.544	3.63	
	3	3.17	4.334	5.273	6.018	6.613	7.101	7.51	7.86	8.163	
	4	4	5.664	7.129	8.4	9.483	10.415	11.226	11.94	12.573	
	5	4.644	6.691	8.565	10.267	11.788	13.145	14.361	15.458	16.455	
	6	5.17	7.523	9.725	11.778	13.666	15.395	16.979	18.436	19.782	
	7	5.615	8.221	10.697	13.04	15.235	17.283	19.19	20.969	22.634	
	8	6	8.824	11.53	14.12	16.577	18.898	21.097	23.152	25.102	
	9	6.34	9.352	12.26	15.061	17.745	20.305	22.74	25.057	27.263	

Noted that when the number of middle forwarding nodes m = 2, since p is always equal to 0.5, we can get the expression of Hdmax about the number of forwarding times N as

Hdmax(N)=N+1−12N∑i=0NCNilog2⁡CNi≈12log2⁡N+2

which shows that the impact of forwarding times on end-to-end delay is logarithmic.

Randomness analysis of end-to-end delay in equal delay forwarding network

We have known that collision leads to the decrease of the randomness of end-to-end delay in RFNs and the Equal Delay Forwarding Network (EDFN) is the most collision-prone network theoretically, which is worth some analysis.

EDFN is defined as a forwarding network, in which the delay between nodes is approximately the same. In EDFN, for any node Zi, there is no difference between forwarding to Zj1 or to Zj2. From the symbolic point of view, Zj1 and Zj2 can rotate. As shown in Fig. 9, let p denotes the self-forwarding probability of middle forwarding nodes and q denotes the forwarding probability between middle forwarding nodes.

Figure 9 Optimal forwarding strategy for EDFN.

For convenience, the delay between middle forwarding nodes is normalized to 1, then the forwarding matrix P of EDFN is

P=(pqx…qxqxp…qx…………qxqx…p)=(qx)11T+(p−qx)I

where I is the identity matrix and 1 is the m dimensional vector of ones.

Therefore, for the EDFN with m nodes and N forwarding times, the end-to-end delay distribution polynomial P(x) is

p(x)=1m1TPN1=(1m1TP1)N=(p+(m−1)qx)N

That is to say, the end-to-end delay of EDFN obeys binomial distribution, and the maximum entropy of binomial distribution is obtained at p = 0.5, so the optimal forwarding strategy for EDFN is

{p=0.5q=12(m−1)

Then the end-to-end delay distribution polynomial p(x) under the optimal forwarding strategy is

p(x)=12N(1+x)N=12N∑i=0NCNixi

So the end-to-end delay distribution of EDFN under the optimal forwarding strategy is p(d=i)=12NCNi, and the maximum entropy of the end-to-end delay in EDFN is

Hdmax=N−12N∑i=0NCNilog2⁡CNi

It can be found that the maximum entropy of EDFN is only related to the number of forwarding times N, and is irrelevant to the number of middle forwarding nodes m. What’s worse, the maximum entropy of EDFN is 1 bit lower than the maximum entropy of RFNs with 2 nodes under the inevitable collision deployment. So, this phenomenon also strongly proves the conclusion that collision is the main reason for the decrease of randomness.

Application: using the randomness of end-to-end delay to generate symmetric keys

Key generation needs random sources. The original key distribution channel tends to be insecure, so the original key exchange is a difficult problem. One idea is using the key distribution center (KDC) to generate random numbers and then realize the key distribution through the secure key exchange protocol (D’Arco, 2001). In 1976, Diffie & Hellman (1976) proposed a key exchange scheme using discrete logarithm, but there is also a man-in-the-middle attack problem, and the security is dependent on the NP problem of discrete logarithm in the finite field on classical computers. The development of quantum computing has impacted the cryptography algorithm based on discrete logarithm problems. Shor (1999) has proved that there exist polynomial-time algorithms for prime factorization and discrete logarithms on a quantum computer.

Another way of thinking is to abandon the idea that the key is distributed by the center, and choose the scheme that both sides of the communication measure the channel to obtain reciprocity characteristics. This process does not need secret information exchange, so it avoids the risk that secret information leaks. For example, the key is generated by using the frequency selective fading characteristic of the wireless channel, including measuring the received signal strength (RSS) (Awan et al., 2019), the channel impulse response (CIR) in time-frequency domain (Walther, Franz & Strufe, 2019), and the phase (Zeinali & Khaleghi Bizaki, 2016), delay and envelope of the received channel (Ye et al., 2010). The only problem is that the spatial distance between sender and receiver is limited in wireless channel key exchange, and the information exchange is mainly carried out by wire for the equipment with a far geographical distance. There is also a lot of randomness in RFNs, and the end-to-end delay, which is mainly studied in this paper, is an ideal feature that satisfies both long-term randomness and short-term reciprocity and can be used for key generation. So, this section mainly introduces how to use the randomness of end-to-end delay to generate symmetric keys.

As shown in Fig. 10, the whole process of symmetric key generation based includes RFNs deployment, forwarding strategy setting, secure measurement of end-to-end delay, quantization encoding, and information reconciliation. Each part is described in detail below.

Figure 10 Key generation process based on RFNs.

RFNs deployment

RFNs can be applied in many scenarios, such as the large scenario of host group distributed between cities, or the small scenario of communication node cluster within the scope of LAN, especially in the scenario of encrypted communication needs among IoT device clusters. It is very convenient to generate the symmetric key with end-to-end delay. The deployment of RFNs mainly concerns two indicators, one is the number of middle forwarding nodes, the other is whether there is an evitable collision. The former affects the deployment cost, while the latter affects the efficiency of key generation.

The number of middle forwarding nodes is determined by the demand of the real scene key generation rate. From the perspective of the economy, we hope to achieve the highest key generation rate with the least number of nodes. For example, if the key generation rate of r = 128 bit/s is required, then suppose the average time t¯ required for a single measurement is 100 ms, a single measurement must generate at least 12.8 bit key. From the data in Table 2, when the number of middle forwarding nodes m = 5 and the number of forwarding times N = 6, the key length is 13.145 bits, which can meet the requirement. That is to say, the key length is determined by Hdt¯>r, and the number of middle forwarding nodes is determined by looking up Table 2.

The evitable collision can be checked by calculating p(x). The number of inevitable collisions can be obtained by calculating the symbol matrix SmN and counting the coefficients, and other like terms are all evitable collisions. These evitable collisions can be avoided as far as possible by adjusting the deployment.

Forwarding strategy setting

When the RFNs network is deployed, the optimal forwarding strategy p can be found through Table 1, and then the internode forwarding probability q can be calculated by 1−pm−1. For the above example, the optimal forwarding strategy is p = 0.086 and q = 0.2285 for (m = 5, N = 6). Because of the rotation among nodes, the forwarding strategies set by each node are the same, which is also very helpful in security, because attackers cannot identify forwarding nodes by counting forwarding rules. Although the forwarding strategy seems to be static, the dynamically adjusted forwarding strategy often divulges the information of the network itself, so that attackers can take advantage of it. When the number of forwarding nodes or forwarding times changes, the forwarding strategy of deployed nodes can be easily switched by looking up Table 1.

Secure measurement of end-to-end delay

The consistency of generated keys depends on the accurate measurement of end-to-end delay (Fabini & Abmayer, 2013). In order to ensure that both sides of the communication can measure approximately the same end-to-end delay and meet the security requirements, we design a secure end-to-end delay measurement scheme as shown in Fig. 11. The scheme steps are as follows:

Figure 11 Secure end-to-end delay measurement scheme.

Alice sends a request message to Bob and records the sending time Tab1,

Bob receives the request and records the receiving time Tba1, and send the reply package to Alice with a delay of εB,

Alice receives the reply and sent it to Bob with a delay of εA. Then record the receiving time Tab2, calculate the data transmission delay ΔTab,

Bob receives the reply and records the receiving time Tba2, calculate the data transmission delay ΔTba.

Let dAB denotes the end-to-end delay from Alice to Bob and dBA denotes the end-to-end delay from Bob to Alice. Then according to this scheme, Alice and Bob can calculate ΔTab and ΔTba as measurement end-to-end delay as

ΔTab=Tab2−Tab1=dAB+dBA+εA+εB

ΔTba=Tba2−Tba1=dAB+dBA+εA+εB

Since ΔTab = ΔTba, the end-to-end delays measured by Alice and Bob are equal.

In terms of security, because each node only records the last hop node, Alice is anonymous in the forwarding packet, and only Bob’s information is in the forwarding packet, so it is impossible to measure the end-to-end delay directly from the sending and receiving nodes. It is also difficult to obtain the end-to-end delay by obtaining the forwarding route. Because the forwarding strategy is random, the probability of each node in the next hop is the same, so it cannot be traced. To obtain a complete forwarding route, the attacker needs to attack almost all forwarding nodes, which means that the cost of the attack is far greater than the benefit. So in general, the security of the scheme is guaranteed.

Quantization encoding

When we get the end-to-end delay data, we need to use quantization coding technology to convert it into a key. We use nonlinear quantization, and the distribution of quantization interval is consistent with that of end-to-end delay. Gray code is used in coding because Gray code belongs to reliability coding, which is an error minimization coding method (Mecklenburg, Pehlert & Sullivan, 1973). Another scheme is to encode the distribution of end-to-end delay by Huffman coding (Huffman, 1952), and then make the nearest neighbor decision on the measured end-to-end delay and the theoretically calculated possible value.

Information reconciliation

An information reconciliation protocol is used to discard or correct the difference of key bits generated by the sender and the receiver, which is a common method for key agreement in physical layer security. Existing information reconciliation methods are mainly divided into reconciliation protocols and error correction coding. The reconciliation protocols mainly include BBBSS, Cascade, and Winnow protocol. Error correction coding includes Hamming code, BCH code, Turbo code, LDPC code, etc. (Huth et al., 2016). Of course, if the process of information reconciliation causes information leakage, then privacy amplification is needed to discard some leaked bits (Maurer & Wolf, 2003).

In Purple Mountain Laboratory of Nanjing, we design a symmetric key generation system according to the application introduced in this section (Huang et al., 2021). The practical results show that this scheme is effective. According to our statistics, the key agreement rate of sending and receiving can be over 91%, which can meet our communication needs.

Conclusions

This paper studies the randomness of end-to-end delay in random forwarding networks (RFNs) through the problem of the drunk man returning home. In this paper, we solved six problems in the study of end-to-end randomness in RFNs. By establishing a mathematical model, we solved the first problem of what kind of distribution does end-to-end delay obey by deriving the formula Eq. (1) for calculating the random distribution of end-to-end delay; Then, the second question of how to measure the randomness of end-to-end delay was answered by analyzing the end-to-end delay generation model, and the conclusion is that the randomness of end-to-end delay can be quantitatively measured by information entropy; In the process of answering the third question of what is the reason for decline of the randomness of end-to-end delay, we found that the end-to-end delay collision is the main reason, among which the evitable collision can be solved by adjusting RFNs deployment, while the inevitable collision can not be avoided; Then, we proposed a fast algorithm FPSSM (Algorithm 1) for calculating inevitable collisions by using symbolic matrix and solved the optimization problem of maximizing the randomness of end-to-end delay to answer the fourth and fifth questions of what is the upper limit of end-to-end delay and how to reach the upper limit. We gave the flow of solving the optimization problem in detail, and then gave the optimization results in Table 1: the upper limit of the randomness of end-to-end delay and Table 2: the optimal forwarding strategy; Finally, we introduced the application of symmetric key generation based on end-to-end delay randomness to answer the final question of how to use the RFNs to share keys.

Supplemental Information

Supplemental Information 1 Proof of the formula in our paper.

Click here for additional data file.

Supplemental Information 2 The end-to-end delay data.

Click here for additional data file.

Supplemental Information 3 The experimental code.

Click here for additional data file.

The authors would like to thank our supervisor Prof. Huang for this interesting research direction, the research environment of Purple Mountain Laboratory, and the lab team for their great help.

Additional Information and Declarations

Competing Interests

Author Contributions

Data Availability

The authors declare that they have no competing interests.

Xiaowen Wang conceived and designed the experiments, performed the experiments, analyzed the data, performed the computation work, prepared figures and/or tables, authored or reviewed drafts of the paper, and approved the final draft.

Jie Huang conceived and designed the experiments, analyzed the data, authored or reviewed drafts of the paper, and approved the final draft.

Zhenyu Duan conceived and designed the experiments, performed the experiments, analyzed the data, performed the computation work, prepared figures and/or tables, and approved the final draft.

Yao Xu performed the experiments, performed the computation work, prepared figures and/or tables, and approved the final draft.

Yifei Yao performed the experiments, performed the computation work, prepared figures and/or tables, and approved the final draft.

The following information was supplied regarding data availability:

The complete code for all of the experimental results is available in the Supplemental File and at GitHub: https://github.com/wxwled/RFN-project.

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
