# Peer review of "Randomness analysis of end-to-end delay in random forwarding networks"

_PeerJ Computer Science, doi:10.7717/peerj-cs.942_

## Round 0.1 · original submission · Minor Revisions

The authors presented a better work by proposing a mathematical model of random forwarding networks (FNs) and derive the expression of end-to-end delay distribution in different FNs. However, the work require improvements, as the reviewers recommended. We advise you to please kindly consider all recommended comments and concerns:
1. Better to rephrase your abstract and add the results comparison with the existing benchmark and elaborate more on the contribution of your presented work.
2. Be more specific and clearer about your system model working in different case scenarios.
3. A thorough proofread is recommended to avoid typos and improve the quality of your manuscript.
4. Please double check for the information provided with the references, a few of references require attention. Also, if possible add a few more related references as well.
5. Better to make the figures captions more descriptive.

Reviewer 1 ·

Basic reporting

In Line 12 in the abstract: both receiver and receiver
I believe this is an unforgivable mistake, please modify it
The introduction isn't written well
This manuscript needs substantial copyediting and English writing revisions
Literature references are insufficiently provided.

Experimental design

No comment

Validity of the findings

No comment

Reviewer 2 ·

Basic reporting

The authors propose a mathematical model of random forwarding networks (FNs) and derive the expression of end-to-end delay distribution in different FNs. The topic is interesting, but the quality of the manuscript can be improved in terms of its problem novelty and main contribution. The following comments are provided for the authors’ consideration:
1. Please try to demonstrate more results in comparing different parameter settings and benchmarks. It would be better that some comparisons between existing works and the proposed algorithm are provided.
2. Regarding the system model part, is the model specified for only one sender and one receiver? What is the adjustment if we increase the number of senders and receivers?
3. The abstract should be revised. For instance, “As a random quantity easily obtained by both receiver and receiver …” should be replaced by “As a random quantity easily obtained by both sender and receiver”.
4. Some references are wrongly referred. For example, Zhang, J., Li, G., Marshall, A., Hu, A., and Hanzo, L. (2020). A new frontier for iot security emerging from three decades of key generation relying on wireless channels. IEEE Access, PP(99) should be corrected. In fact, the page numbers is 138406-138446 and the volume is 8. Please check that.
5. Some equations are not numbered.
6. The caption of Figure 8 should be revised (H_d as a function of p for m = 3 and N = 2; and put H_d (bits) as ylabel).

Experimental design

Regarding the system model part, is the model specified for only one sender and one receiver? What is the adjustment if we increase the number of senders and receivers?

Validity of the findings

Please try to demonstrate more results in comparing different parameter settings and benchmarks. It would be better that some comparisons between existing works and the proposed algorithm are provided.

---

## Round 0.2 · accepted · Accept

Congratulations on acceptance. However, it is recommended to have a thorough proofread.

Reviewer 1 ·

Basic reporting

the authors have addressed all the mentioned comments, i have no more comments

Experimental design

The authors have addressed all the mentioned comments, i have no more comments

Validity of the findings

The authors have addressed all the mentioned comments, i have no more comments

Additional comments

The authors have addressed all the mentioned comments, i have no more comments